# Effects of Exogenous Salicylic Acid Application to Aboveground Part on the Defense Responses in Bt (*Bacillus thuringiensis*) and Non-Bt Corn (*Zea mays* L.) Seedlings

**DOI:** 10.3390/plants11162162

**Published:** 2022-08-20

**Authors:** Yuanjiao Feng, Xiaoyi Wang, Tiantian Du, Yinghua Shu, Fengxiao Tan, Jianwu Wang

**Affiliations:** 1Key Laboratory of Agro-Environment in the Tropics, Ministry of Agriculture, South China Agricultural University, Guangzhou 510642, China; 2Guangdong Provincial Key Laboratory of Eco-Circular Agriculture, South China Agricultural University, Guangzhou 510642, China; 3Guangdong Engineering Research Center for Modern Eco-Agriculture and Circular Agriculture, Guangzhou 510642, China; 4Department of Ecology, College of Natural Resources and Environment, South China Agricultural University, Guangzhou 510642, China

**Keywords:** Bt corn, salicylic acid, defense responses, Bt protein, defense chemicals, defense enzymes

## Abstract

Bt (*Bacillus thuringiensis*) corn is one of the top three large-scale commercialized anti-insect transgenic crops around the world. In the present study, we tested the Bt protein content, defense chemicals contents, and defense enzyme activities in both the leaves and roots of Bt corn varieties 5422Bt1 and 5422CBCL, as well as their conventional corn 5422 seedlings, with two fully expanded leaves which had been treated with 2.5 mM exogenous salicylic acid (SA) to the aboveground part for 24 h. The result showed that the SA treatment to the aboveground part could significantly increase the polyphenol oxidase activity of conventional corn 5422, the Bt protein content, and peroxidase activities of Bt corn 5422Bt1, as well as the polyphenol oxidase and peroxidase activity of Bt corn 5422CBCL in the leaves. In the roots, the polyphenol oxidase and peroxidase activity of conventional corn 5422, the polyphenol oxidase and superoxide dismutase activities of Bt corn 5422Bt1, the DIMBOA (2,4-dihydroxy-7-methoxy-2H, 1, 4-benzoxazin-3 (4H)-one) content, and four defense enzymes activities of Bt corn 5422CBCL were systematically increased. These findings suggest that the direct effect of SA application to aboveground part on the leaf defense responses in Bt corn 5422CBCL is stronger than that in non-Bt corn. Meanwhile, the systemic effect of SA on the root defense responses in Bt corn 5422CBCL is stronger than that in conventional corn 5422 and Bt corn 5422Bt1. It can be concluded that the Bt gene introduction and endogenous chemical defense responses of corns act synergistically during the SA-induced defense processes to the aboveground part. Different transformation events affected the root defense response when the SA treatment was applied to the aboveground part.

## 1. Introduction

Genetically engineered crops have been cultivated globally for 25 years over a continuously growing area. According to the ISAAA (International Service for the Acquisition of Agri-Biotech Applications), planting areas for transgenic crops have increased significantly, amounting to 190.4 million hectares in 2019 [1]. Meanwhile, the worldwide surface of insect-resistant Bt crops, alone or in combination with other traits such as herbicide tolerance, has exceeded 107 million hectares, among which Bt corn was planted over approximately 66 million hectares worldwide [1]. Corn (*Zea mays* L.) is a major food crop in China [2]. Bt corn is one of the top three large-scale commercialized anti-insect transgenic crops around the world, genetically modified to express insecticidal crystal protein from the bacterium *Bacillus thuringiensis*, which is toxic to some lepidopteran pests [3,4,5,6]. In addition to paying attention to the temporal and spatial expression of Bt protein in Bt corn [2,7,8] and the detection of insect resistance [5,9,10,11,12,13], the ecological risk assessment of Bt corn has also attracted the extensive attention of scholars at home and abroad [14,15,16,17,18]. To date, many studies have investigated the ecological risk assessment of Bt corn, mostly on the effect of Bt corn plantation on nontarget organisms on the ground [17,19,20]. In addition, studies have assessed the impact of Bt corn plantation and straw returning on the soil ecosystem [21,22,23,24,25,26,27].

Corn has evolved several strategies to avoid or defend the diseases and pests during their long coevolution processes, called “induced defense responses” [3,28,29,30,31,32,33]. The ways of inducing defense response usually include mechanical wounding, insect feeding, exogenous jasmonic acid (JA), and salicylic acid (SA) treatment, etc. Such defense responses can be studied by determining the contents of defense chemicals (such as DIMBOA (2,4-dihydroxy-7-methoxy-2H, 1, 4-benzoxazin-3 (4H)-one) and total phenolics) and the activities of some defensive enzymes (such as polyphenol oxidase, peroxidase, catalase, and superoxide dismutase), etc. [4,34]. As we know, the induced defense response in corn is characterized by a systematic response, and the treatment to the leaves may systematically induce the chemical defense response in the roots of corn [31,32]. It is believed that pest control can be achieved by combining the endogenous defense system with the introduced Bt genes, which is a promising alternative strategy for pest resistance management [3,4,35,36,37]. Nowadays, evaluating the difference in the induced defense responses between Bt corn and conventional corn is an important component to assess the ecological risk of transgenic crops [3,4,38,39,40]. So far, the main studies on the induced defense response of Bt corn include mechanical wounding, insect feeding, and exogenous JA treatment, and there are no reports on the direct and systematic effects of exogenous SA treatment on the defense response of Bt corn.

SA is a key signal compound in the induced defense response, hypersensitive response, and in systemic acquired resistance to pathogens in corn, and it can be transferred within the whole plant [31,41,42,43]. A mass of studies have shown that the treatment of exogenous SA to the aboveground part can affect the defense response [44,45,46,47,48,49]. For instance, foliar spraying of SA in *Thymus vulgaris* L. increased the contents of total phenolics and total flavonoids in the leaves after two months postapplication [44]. In the present study, after the aboveground part of Bt corns (5422Bt1 and 5422CBCL) and their conventional corn 5422 with two fully expanded leaves were treated with 2.5 mM exogenous SA for 24 h, we examined the content of Bt protein, DIMBOA, total phenolics, and the activities of polyphenol oxidase, peroxidase, catalase, and superoxide dismutase in the leaves and roots. It aimed to compare the direct induced defense response in the leaves and the systematic induced defense response in the roots of Bt corns and non-Bt corn with the application of SA to aboveground part. The findings of our study can be helpful in understanding the defense responses in Bt crops against diseases and pests, and also provide useful information for the assessment of the ecological risk of transgenic Bt crops.

## 2. Results

### 2.1. Direct and Systemic Effect of Exogenous Salicylic Acid Application to the Aboveground Part on the Content of Defense Chemicals in the Leaves and Roots of Bt and Non-Bt Corn Seedlings

#### 2.1.1. Bt Protein Content

SA application to aboveground part significantly increased the content of Bt protein in the leaves of Bt corn 5422Bt1 by 59.12% (*t* = 3.492, *p* = 0.013), but had no obvious direct effect on the content of Bt protein in the leaves of another Bt corn, 5422CBCL (Figure 1).

The aboveground part treated with SA did not systematically effect the Bt protein content in the roots of the two Bt corns (5422Bt1 and 5422CBCL) (Figure 1).

#### 2.1.2. DIMBOA Content

There was no difference in the DIMBOA content in the leaves of the conventional corn and the Bt corns (5422Bt1 and 5422CBCL) after the application of SA to the aboveground part (Figure 2).

The SA application to aboveground part had no significant systemic effect on the DIMBOA content in the roots of the conventional corn 5422 and the Bt corn 5422Bt1 (Figure 2), but it increased systematically the DIMBOA content in the roots of another Bt corn, 5422CBCL, by 573.63% (*t* = 4.521, *p* = 0.004).

#### 2.1.3. Total Phenolic Content

The SA application to aboveground part had no significant direct effect on the content of the total phenolics in the leaves of the two Bt corns and the conventional corn (Figure 3).

The SA treatment of the aboveground part had no significant systematic effect on the content of the total phenols in the roots of the two Bt corns (5422Bt1 and 5422CBCL) and the conventional corn 5422 (Figure 3).

### 2.2. Direct and Systemic Effect of Exogenous Salicylic Acid Application to the Aboveground Part on the Activity of Defense Enzymes in the Leaves and Roots of Bt and Non-Bt Corn Seedlings

#### 2.2.1. Polyphenol Oxidase Activity

The direct effect of the SA treatment to the aboveground part on the activity of polyphenol oxidase of the conventional corn and the Bt corns in the leaves was different (Figure 4). The SA treatment of the aboveground part increased significantly the polyphenol oxidase activity in the leaves of the conventional corn 5422 by 74.28% (*t* = 2.903, *p* = 0.027), and that of the Bt corn 5422CBCL leaves by 135.58% (*t* = 6.545, *p* = 0.001), but it had no remarkable direct effect on the polyphenol oxidase activity in the leaves of the other Bt corn.

The SA application to the aboveground part had a significant systematic effect on the activity of polyphenol oxidase in the roots of the conventional corn 5422 and the two Bt corns (Figure 4). After the SA was applied to the aboveground part, the polyphenol oxidase activity in the roots of the conventional corn and the Bt corns (5422Bt1 and 5422CBCL) was systematically increased by 68.33% (*t* = 3.802, *p* = 0.009), 62.59% (*t* = 4.649, *p* = 0.004), and 92.15% (*t* = 4.543, *p* = 0.004), respectively.

#### 2.2.2. Peroxidase Activity

The SA application to the aboveground part increased tremendously the peroxidase activity in the leaves of the Bt corn 5422Bt1 and 5422CBCL by 94.06% (*t* = 2.758, *p* = 0.033) and 74.04% (*t* = 2.629, *p* = 0.039), respectively; however, it had no significant direct effect on the peroxidase activity in the leaves of the conventional corn 5422 (Figure 5).

The SA treatment to the aboveground part systematically increased the peroxidase activity in the roots of the conventional corn 5422 and Bt corn 5422CBCL by 105.11% (*t* = 5.309, *p* = 0.010) and 48.25% (*t* = 3.955, *p* = 0.022), respectively, but it had no obvious systematic effect on the peroxidase activity in the roots of another Bt corn, 5422Bt1 (Figure 5).

#### 2.2.3. Catalase Activity

The SA treatment to the aboveground part had no obvious direct effect on the content of the catalase activity in the leaves of the two Bt corns (5422Bt1 and 5422CBCL) and the conventional corn (Figure 6).

The SA application to the aboveground part had no significant systematic effect on the activity of the catalase in the roots of the conventional corn 5422 and the Bt corn 5422Bt1 (Figure 6) but increased systematically the catalase activity in the roots of the Bt corn 5422CBCL by 335.54% (*t* = 4.880, *p* = 0.011).

#### 2.2.4. Superoxide Dismutase Activity

There was no difference in the activity of superoxide dismutase in the leaves of the conventional corn and the Bt corns after the application of the SA to the aboveground part (Figure 7).

The salicylic acid treatment to the aboveground part had no obvious systematic effect on the superoxide dismutase activity in the roots of the conventional maize 5422 (Figure 7), but it increased systematically the superoxide dismutase activity in the roots of the Bt corns (5422Bt1 and 5422CBCL) by 129.63% (*t* = 2.762, *p* = 0.033) and 135.47% (*t* = 3.206, *p* = 0.018), respectively.

## 3. Discussion

SA is a ubiquitous endogenous signal molecule in plants that can activate the defense and protection mechanisms for disease resistance, and it plays a critical role in plant allergic reactions to pathogens and systemic acquired resistance [31,43]. A mass of studies had proved that the treatment of exogenous SA to the aboveground part can affect directly the foliar defense response [31,44,45,46,47,48,49]. For example, SA foliage application further enhanced the activities of enzymes (SOD, POD, and CAT) and nonenzymatic antioxidants such as ascorbic acid and the phenolic contents [46]. Our research also showed that the SA treatment to the aboveground part could significantly increase the polyphenol oxidase activity of conventional corn 5422, the Bt protein content and peroxidase activities of Bt corn 5422Bt1, and the polyphenol oxidase and peroxidase activity of Bt corn 5422CBCL in the leaves.

Our study verified that SA application to the aboveground part significantly increased the content of Bt protein in the leaves of the Bt corn 5422Bt1, and this was similar to previous studies. For instance, the Bt protein content was increased in the first leaf of the Bt corn 34B24 by being treated with JA to the first leaf [37]. However, some studies showed that mechanical wounding, Asian corn borer (*Ostrinia furnacalis*) feeding, and exogenous JA treatment to the aboveground part had no significant effect on the Bt protein content in the leaves of the Bt corn 5422Bt1 and 5422CBCL [3,38,39]. In addition, our research showed that the direct effect of SA application to the aboveground part on the leaf defense responses in Bt corn 5422CBCL is stronger than that in non-Bt corn. For example, the SA application to the aboveground part increased tremendously the peroxidase activity in the leaves of the Bt corn 5422CBCL; however, it had no significant direct effect on the peroxidase activity in the leaves of the conventional corn 5422. It can be concluded that the Bt gene introduction and endogenous chemical direct defense responses in the leaves of corn act synergistically during the SA-induced defense processes to the aboveground part. These conclusions are consistent with the previous research results [3,38,39].

As we know, the induced defense response in plants is characterized by a systematic response [31,50,51]. Studies have reported that exogenous SA treatment to the aboveground part can affect systematically the defense response of roots [31,52,53,54,55]. For example, Khanna et al. [54] found that the root activities of CAT and SOD increased by 400.0 μM SA within 2 h of foliar treatment with SA in corn seedlings. Similarly, foliar application of 1.0 mM SA in the Oueslati olive variety (*Olea europeae* L.) resulted in increased root contents of total phenolics and flavonoids after 15 d [55]. Our results also proved the polyphenol oxidase and peroxidase activity of conventional corn 5422, the polyphenol oxidase and superoxide dismutase activities of Bt corn 5422Bt1, the DIMBOA content, and the four defense enzymes activities of Bt corn 5422CBCL in the roots were systematically increased after the SA treatment to the aboveground part.

In the present study, the aboveground part treated with SA did not systematically effect the Bt protein content in the roots of the two Bt corns (5422Bt1 and 5422CBCL). The previous studies also showed no significance in the Bt protein content in the roots of Bt corn 5422Bt1 and 5422CBCL after mechanical wounding and exogenous JA treatment to the aboveground part [3,38]. However, the Bt protein content was increased systematically in the roots of the Bt corn 5422CBCL which had been damaged by Asian corn borer in the first leaf [39]. Meanwhile, our study showed that the systemic effect of SA application to the aboveground part on the root defense responses in the Bt corn 5422CBCL is stronger than that in the non-Bt corn. For instance, the SA application to the aboveground part increased significantly the DIMBOA content, catalase, and superoxide dismutase activity in the roots of the Bt corn 5422CBCL; however, it had no significant systemic effect in the roots of the conventional corn 5422. It also can be concluded that the Bt gene introduction and the endogenous chemical systemic defense responses in the roots of the corns act synergistically during the SA-induced defense processes to the aboveground part. These conclusions are consistent with the previous research results [3,38]. However, the systematically induced chemical defense response was weaker in the roots for the Bt corns than the conventional corn after the first leaf was damaged by the Asian corn, which implies an antagonistic relationship between the Bt gene introduction and the chemical defense response in the roots [39].

Although the two Bt maize expressed the same insecticidal protein, their conversion events were different. The transformation event of Bt corn 5422Bt1 is Bt11, and the other Bt corn 5422CBCL is Mon810. Our results indicated that the SA treatment to the aboveground part could significantly increase the Bt protein content and peroxidase activities of Bt corn 5422Bt1, and the polyphenol oxidase and peroxidase activity of the Bt corn 5422CBCL in the leaves. However, in the roots, the polyphenol oxidase and superoxide dismutase activities of the Bt corn 5422Bt1, the DIMBOA content, and the four defense enzymes activities of the Bt corn 5422CBCL were systematically increased. These findings suggest that that the systemic effect of the SA application to the aboveground part on the root defense responses in the Bt corn 5422CBCL is stronger than that in the other Bt corn 5422Bt1. It can be concluded that different transformation events affected the root defense response when the SA treatment was applied to the aboveground part, and this was similar to previous studies [3,38,39]. For example, the gene expression of *Bx6*, *Bx9*, *PAL*, *PR-2a*, and *TPS* in the roots of the Bt corn 5422CBCL (Mon810) could be systematically induced by pest damage, whereas only the gene expressions of *Bx9*, *PAL*, and *TPS* in the roots of the Bt corn 5422Bt1 (Bt11) could be systematically induced by pest damage [39]. These results may be due to the content of the chemical defense substances of the corn itself, effected by different transformation events, which leads to the difference in their induced defense responses. The specific mechanism needs to be further studied.

It is well-known that the aboveground and belowground parts of plants are connected closely by the xylem and phloem vessels through which water, nutrients, photosynthates, and other plant compounds are transported [50,56]. Hence, the signal compounds and defense chemicals induced by disease and pests, or exogenous chemicals in the belowground parts may be transmitted to the aboveground parts, and accordingly, initiate the corresponding defense responses in the aboveground parts. Therefore, the research in regard to the induced defense responses of plants belowground has attracted wide attention from scholars [4,31,32,33,57,58,59]. Wang et al. [4] indicated that the Bt gene introduction affects the induced defense effects of the JA treatment to the belowground part of corn, leading to a stronger defense response in the roots of the Bt corns than in the conventional corn. Several studies were carried out to understand the systemic responses in nontreated leaves by exogenous SA treatment to the belowground part [31,57,58,59]. For instance, Song et al. [58] showed that the activities of POD, CAT, and SOD in the leaves were enhanced distinctly when the roots were treated with 0.1 mM SA after 21 d. As a result, it is still necessary to investigate the differences of the direct or systemically induced defense in the leaves and roots of Bt corn and non-Bt corn after the application of SA to the belowground parts.

## 4. Materials and Methods

### 4.1. Materials

Two varieties of Bt corn plants expressing insecticidal Cry1Ab protein, which were donated by Cindy Nakatsu (Department of Agriculture, Purdue University) and provided by Beck’s Hybrids Company (Atlanta, IN, USA), were used for experiments. They were Bt corns 5422Bt1 (Bt11) and 5422CBCL (Mon810), and their conventional corn 5422. The transformation events of two of Bt corns were different. The transformation event of Bt corn 5422Bt1 is Bt11 and for Bt corn 5422CBCL it is Mon810. SA ((±)-Salicylic acid) was purchased from the Sigma-Aldrich company (St. Louis, MI, USA). The molecular weight of solid SA is 138.12 g·mol^−1^ (99%).

### 4.2. Experimental Design

Corn seeds were surface-sterilized in 5% hydrogen peroxide solution for 5 min and then germinated in wet cheese cloth at 25 ± 1 °C. Corn plants were grown in nutrient solution and kept at 22–28 °C under a light: dark 12:12 h photoperiod and 70% relative humidity in a growth chamber (Institute of Tropical and Subtropical Ecology, South China Agricultural University). The seedlings were then transferred to a 500 mL nutrient solution (5 mM KNO_3_, 5 mM Ca(NO_3_)_2_, 2 mM MgSO_4_, 1 mM KH_2_PO_4_, 46 μM H_3_BO_3_, 9 μM MnCl_2_, 0.8 μM ZnSO_4_, 0.3 μM CuSO_4_, 0.1 μM H_8_MoN_2_O_4_, 20 μM FeNaEDTA) with two seedlings per pot every other two days [4]. We have examined effects of the salicylic acid (SA) concentration (0.1, 0.5, 1.0, 2.5, 5.0 mM) and time (3, 12, 24, 48, 72 h) on the defense responses in corn after foliar application. The results showed that SA foliar application at 2.5 mM produces strong defense responses in corn, with the optimal induction time at 24 h. Treatments were applied to the corn seedlings with two fully expanded leaves. There were two treatments with four repeats: 2.5 mM salicylic acid treatment (SA) and the control (CON). In the 2.5 mM SA treatment, 100 μL SA solution containing 0.14% ethanol and 0.05% Tween-20 was spread uniformly on two fully expanded leaves of seedlings. The same volume of distilled water with 0.14% ethanol and 0.05% Tween-20 was used as control. After the leaves were treated for 24 h, the leaves (the treated part) and roots (the nontreated part) of each plant were collected separately to determine the content of Bt protein, DIMBOA, total phenolics, and the activities of polyphenol oxidase, peroxidase, catalase, and superoxide dismutase.

### 4.3. Analysis of Bt Protein

Bt protein in the leaf and root was quantified using commercial enzyme-linked immunosorbent assay (ELISA) kits according to the protocol of manufacturer (Agdia Company, Elkhart, IN, USA). Briefly [38], a 20 mg leaf sample or a 30 mg root sample was ground into powder in liquid nitrogen and immediately transferred into 2 mL centrifugal tube. The samples were then extracted by 1 mL PBST (Phosphate-Buffered Saline-Tween, provided with the kit). They were mixed thoroughly and centrifuged under 12,000 rpm at 4 °C for 10 min. The supernatant was diluted at a certain ratio (leaf: 1000:1; root: 600:1) with PBST for further detection. Diluted sample of 100 μL was added into each well of the plate (supplied with the kit), shaken for 15 min at 200 r·min^−1^ after covering preservative film, and followed by a 2 h incubation at room temperature. Then, the plate was washed 5 times with PBST and incubated for another 2 h after adding 100 μL enzyme conjugate in each well. The washing was repeated, and then 100 μL of TMB (Tetramethylbenzidine) substrate was added. After 15 min incubation, 50 μL 3 M H_2_SO_4_ was added to stop color development. Absorbance was measured at a wavelength 450 nm with a molecular devices microplate reader (Molecular Devices, San Jose, CA, USA) within 30 min. Bt protein concentration was measured by a five-point standard curve of purified Cry1Ab (supplied with the kit).

### 4.4. Analysis of DIMBOA

The procedure to prepare samples for the DIMBOA concentration was slightly modified from Ni and Quisenberry [60]. Fresh leaves and roots were weighed and ground into powder with a mortar in 10 mL distilled water. Aqueous extracts were incubated for 20 min, and samples were diluted with methyl alcohol in a ratio of 1:1. The methanol-diluted extract was centrifuged at 12,000 rpm for 15 min and filtered. The filtrate was evaporated to dryness under a vacuum. The residue was dissolved in 2 mL mixed solution (acetonitrile: 0.5% aqueous acetic acid, 1:1, *v*/*v*). Extracts were filtered through 0.45 μm membrane filters, and then the samples were stored at −20 °C for further measurement.

DIMBOA concentrations in the samples were quantified by high-performance liquid chromatography (HPLC) (Agilent 1100, USA) (column, Hypersil ODS C_18_ column (250 mm × 4 mm, 5 μm)) with a DAD detector by using external standard curves. Gradient elution was performed with a gradient of A (acetonitrile) and B (0.5% aqueous acetic acid), i.e., 25–45% of A from 0–10 min and 45–25% of A from 10–15 min. Solvent flow rate was set at 1 mL·min^−1^. The injection volume was 20 μL and the detection wavelength was 262 nm. DIMBOA contents in leaves and roots were determined according to the standard calibration curve obtained by peak area of a series of concentrations (0, 20, 40, 60, 80, 100 μg·mL^−1^) of DIMBOA standard samples. DIMBOA standard sample was purchased from the Shanghai ACMEC biochemical technology company. The purity of DIMBOA standard sample was 99%.

### 4.5. Analysis of Total Phenolics

Total phenolics contents were assayed according to Randhir and Shetty [61] and were determined as gallic acid equivalents (0, 25, 50, 100, 150, 200 μg·mL^−1^). Samples were weighed and ground into power in liquid nitrogen, soaked in 10 mL of 95% ethanol, and then kept in freezer for 48 h. The sample was centrifuged at 12,000 rpm for 10 min and filtered. Leaf: the filtrate of 0.5 mL was transferred into a test tube, and then 1.5 mL of 95% ethanol, 5 mL of distilled water, and 0.5 mL of Folin–Ciocalteu phenol reagent were added. Root: the filtrate of 1 mL was transferred into a test tube, and then 1 mL of 95% ethanol, 5 mL of distilled water, and 0.5 mL of Folin–Ciocalteu phenol reagent were added. After an incubation period of 5 min, 1 mL of 5% Na_2_CO_3_ was added, mixed well, and kept in dark for 1 h. The content of total phenolics was measured at 725 nm using UV–visible spectrophotometer (UV-2450 SHIMADZU, Kyoto, Japan).

### 4.6. Analysis of Polyphenol Oxidase Activity

Polyphenol oxidase (PPO) crude enzyme preparation was done according to Sivakumar and Sharma [62]. The samples were homogenized individually with 1 mL 0.1 M phosphate buffer (PH 6.5) in the ratio of 1:5 (*w*/*v*), centrifuged at 6000 rpm at 4 °C for 15 min, and the supernatant was used as crude enzyme for estimation.

The crude enzyme solution (10 μL), sample dilution (40 μL), and 6 concentrations (0, 15, 30, 60, 120, 180 U·L^−1^) of standard solutions were added to each well of microplate (Rapidbio Company, Plymouth, MI, USA), and thereafter the wells were incubated for 30 min at 37 °C. The microplate wells were washed with buffer for five times, and 50 μL HRP conjugate reagent was added and the wells, which were incubated and then washed five more times. An amount of 50 μL chromogen solution A and 50 μL chromogen solution B were added to each well and kept in dark for 15 min at 37 °C, then 50 μL stop solution was added to each well. The absorbance at 450 nm was measured by microplate reader. The polyphenol oxidase activity was calculated using a regression equation of the standard curve with the standard density and OD value.

### 4.7. Analysis of Peroxidase Activity

The activity of peroxidase (POD) in the leaf and root was quantified using the guaiacol colorimetric method described by Gao [63]. About 0.1 g of the sample was homogenized with 1 mL of 0.05 mol·L^−1^ PBS (phosphate-buffered saline) in an ice bath and centrifuged at 4 °C for 15 min at 8000 rpm. The supernatant was collected and used for the assay. For POD, the oxidation of guaiacol was measured by the increase in absorbance at 470 nm in every 30 s for 2 min. The assay contained 0.95 mL of 0.2% guaiacol, 1 mL phosphate buffer solution (pH 7.0), and 0.05 mL enzyme extract. The reaction was started with 1 mL of 0.3% H_2_O_2_.

### 4.8. Analysis of Catalase Activity

The activity of catalase (CAT) in the leaf and root was determined by hydrogen peroxide decomposition according to the method of Li [64]. An amount of 0.1 g of the sample was homogenized with 1 mL of 0.05 mol·L^−1^ PBS in an ice bath, then centrifuged at 4 °C for 15 min at 8000 rpm. The supernatant was collected and used for the assay. For CAT, the decomposition of H_2_O_2_ was followed by the decline in absorbance at 240 nm in every 30 s for 2 min. The 3 mL reaction mixture contained 1 mL phosphate buffer solution (pH 7.0), 1 mL of 0.3% H_2_O_2_, 0.95 mL of 0.2% guaiacol, and 0.05 mL enzyme extract. The reaction was initiated with enzyme extract.

### 4.9. Analysis of Superoxide Dismutase Activity

The activity of the superoxide dismutase (SOD) in the leaf and root was determined by measuring its ability to inhibit the photochemical reduction of nitroblue tetrazolium (NBT) according to the method of Gao [63]. An amount of 0.1 g of the sample was homogenized with 1 mL of 0.05 mol·L^−1^ PBS (pH 7.8) in an ice bath, then centrifuged at 4 °C for 15 min at 8000 rpm. The 3 mL reaction mixture contained 1.75 mL of 0.05 mol·L^−1^ PBS (pH 7.8), 0.3 mL of 130 mmol·L^−1^ methionine, 0.3 mL of 750 μmol·L^−1^ NBT, 0.3 mL of 100 μmol·L^−1^ EDTA-Na_2_, and 0.05 mL of the enzyme extract. An amount of 0.3 mL of 20 μmol·L^−1^ riboflavin was added last. The test tubes containing the mixture were placed under two fluorescent lamps at 4000 lux. The reaction was started by switching on the light and was allowed to run for 20 min. The reaction was stopped by switching off the light and the absorbance at 560 nm was recorded. A nonirradiated reaction mixture that did not develop color was used as the control, and its absorbance was subtracted from *A_560_*. The reaction mixture which lacked enzyme developed maximum color was the result of the maximum reduction of NBT. The reduction of NBT was inversely proportional to the enzyme activity. One unit of superoxide dismutase activity was defined as the amount of enzyme required to cause 50% inhibition of the rate of NBT reduction at 560 nm.

### 4.10. Statistical Analysis

The data were expressed as the means ± standard errors of four repeats. Analysis of group *t*-test was carried out using SPSS 13.0 (IBM, Armonk, MI, USA), and the significances were tested at 0.05 level using group *t*-test.

## 5. Conclusions

Our results indicated that direct effect of the SA application to the aboveground part on the leaf defense responses in the Bt corn 5422CBCL was stronger than that in conventional corn 5422. Meanwhile, the systemic effect of the SA application to the aboveground part on the root defense responses in the Bt corn 5422CBCL was stronger than that in the conventional corn 5422 and the Bt corn 5422Bt1. It can be concluded that the Bt gene introduction and the endogenous chemical defense responses of the corns act synergistically during the SA-induced defense processes to the aboveground part. Different transformation events affected the root defense response when the SA treatment was applied to the aboveground part.

## Figures and Tables

**Figure 1 plants-11-02162-f001:**
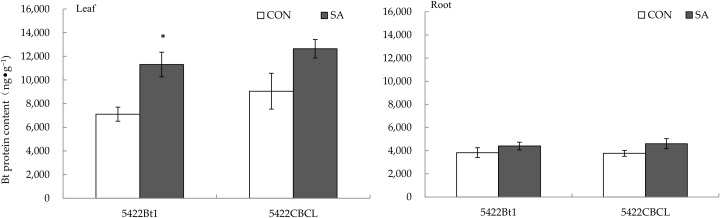
Direct and systemic effects of exogenous salicylic acid application to the aboveground part on the content of Bt protein in the leaves and roots of the Bt corns. The data in the figure are means ± standard errors. Significance among the data was determined by paired Student’s *t*-test. “*” showed *p* < 0.05. CON: control; SA: salicylic acid.

**Figure 2 plants-11-02162-f002:**
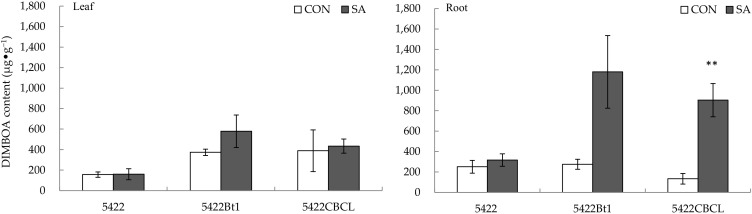
Direct and systemic effects of exogenous salicylic acid application to the aboveground part on the content of DIMBOA in leaves and roots of conventional and Bt corns. The data in the figure are means ± standard errors. Significance among the data was determined by paired Student’s *t*-test. “**” showed *p* < 0.01. CON: control; SA: salicylic acid; DIMBOA: 2,4-dihydroxy-7-methoxy-2H, 1,4-benzoxazin-3 (4H)-one.

**Figure 3 plants-11-02162-f003:**
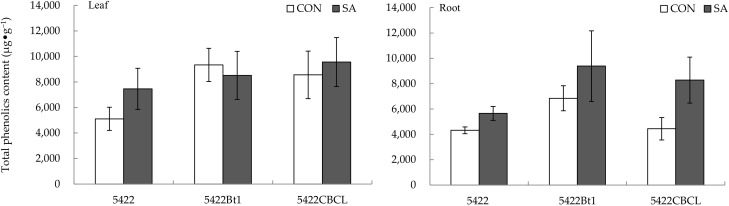
Direct and systemic effects of exogenous salicylic acid application to the aboveground part on the content of total phenolics in leaves and roots of conventional and Bt corns. The data in the figure are means ± standard errors. Significance among the data was determined by paired Student’s *t*-test. CON: control; SA: salicylic acid.

**Figure 4 plants-11-02162-f004:**
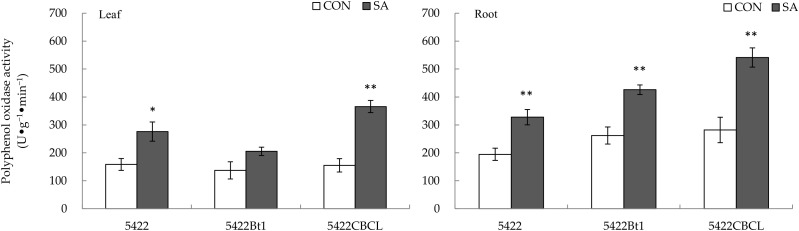
Direct and systemic effects of exogenous salicylic acid application to the aboveground part on the activity of polyphenol oxidase in the leaves and roots of conventional and Bt corns. The data in the figure are means ± standard errors. Significance among the data was determined by paired Student’s *t*-test. “*” showed *p* < 0.05, “**” showed *p* < 0.01. CON: control; SA: salicylic acid.

**Figure 5 plants-11-02162-f005:**
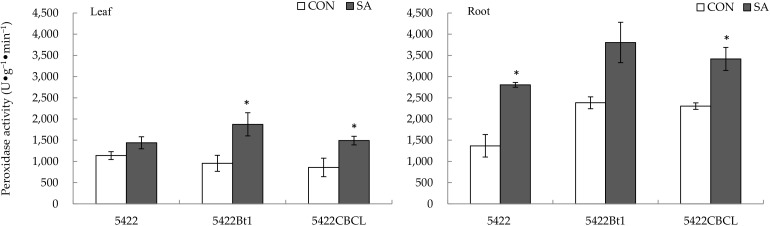
Direct and systemic effects of exogenous salicylic acid application to the aboveground part on the activity of peroxidase in the leaves and roots of conventional and Bt corns. The data in the figure are means ± standard errors. Significance among the data was determined by paired Student’s *t*-test. “*” showed *p* < 0.05. CON: control; SA: salicylic acid.

**Figure 6 plants-11-02162-f006:**
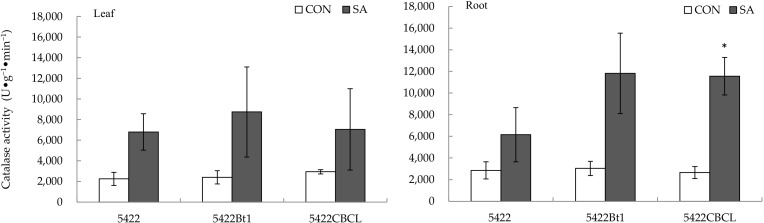
Direct and systemic effects of exogenous salicylic acid application to the aboveground part on the activity of catalase in the leaves and roots of conventional and Bt corns. The data in the figure are means ± standard errors. Significance among the data was determined by paired Student’s *t*-test. “*” showed *p* < 0.05. CON: control; SA: salicylic acid.

**Figure 7 plants-11-02162-f007:**
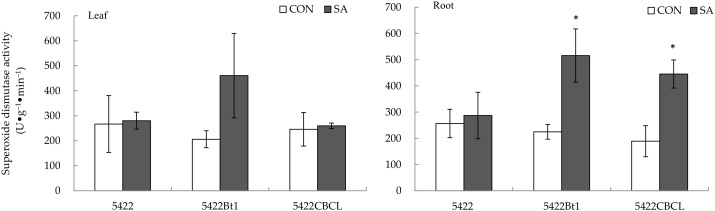
Direct and systemic effects of exogenous salicylic acid application to the aboveground part on the activity of superoxide dismutase in the leaves and roots of the conventional and Bt corns. The data in the figure are means ± standard errors. Significance among the data was determined by paired Student’s *t*-test. “*” showed *p* < 0.05. CON: control; SA: salicylic acid.

## Data Availability

Data recorded in the current study are available in all Tables and Figures of the manuscript.

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
