# Peer review of "Effects of Exogenous Salicylic Acid Application to Aboveground Part on the Defense Responses in Bt (*Bacillus thuringiensis*) and Non-Bt Corn (*Zea mays* L.) Seedlings"

_plants, 2022, doi:10.3390/plants11162162_

Round 1
Reviewer 2 Report
The manuscript under review represents an interesting study about the application of salicylic acid and its relation with the defence system in Zea mays L. plants. However, must be improved.
In the introduction, in the first paragraph, please add information about cultivation and production worldwide to demonstrate its importance of the study.
Line 60: Please, define DIMBOA. Also, in abstract on line 28
It is also necessary to provide more information about the justification for the use of salicylic acid for the determinations. Please, also expand the information, about the mechanisms involved in the defence system of the plant. Also provide more information about what has been reported, in this context, in other crops.
The results are scarce and presented very briefly, separating leaves and roots. In this sense 14 figures has been reported. Please, modify by only two figures, one of the results in leaves (which includes figures 1-7) and another with the results in roots (which includes figures 8- 14).
The statistical treatments should be reviewed. There are cases, where visually, with respect to the error bars, it is evident that there are significant differences between the treatments. However the opposite is reported: Figure 1 5422CBCL, Figure 2 5422Bt1, Figure 3 5422, Figure 5 5422, Figure 6 in all treatments, Figure 7 5422Bt1, Figure 9 5422Bt1, Figure 10 5422Bt1 and 5422CBCL, Figures 12 and 13 5422Bt1.
Figure 1: On the Y axis, please, remove decimals
In all figures, in the corresponding legend, please detail the used abbreviations (CON, SA, 5422Bt1, 5422CBCL, etc)
Figure 13: There is an error in the Y-axis legend. Please, replace peroxidase by catalase.
In points 2.2.5 and 2.2.6, the reported percentages on lines 141 and 142, and 147, do not correspond to what is shown in the respective graphs.
In the discussion, in the last paragraph, it is necessary to add more information, about the reasons for this study. Specifically, about the external agents against which the studied factors act.
In the methodology, in point 4.2, please indicate the reference or reasons for choosing this nutrient solution
Line 323: Please define PBST
Line 345: Please, modify 18 of C18 as a subscript. Also, add the particle size of the column
Lines 351 and 377: Please, add purity and supplier of the standards.
Line 363: Measured at 725nm in a UV spectrophotometer? or UV-Vis?
Line 367: Please, replace 4°Cfor by 4°C for…
Lines 381 and 399: Replace r*min-1 by rpm
Line 399: Please, replace 4°C for 15 min at by 4°C for 15 min at…
Reviewer 3 Report
The manuscript describes effects of salicylic acid (SA) application on the defense responses in two Bt corn plants (5422Bt1 and 5422CBCL). Non Bt corn plant (5422) was used as a control. The results show that SA application improves the defense response of the Bt corn overall.
This reviewer suggests authors describe the insecticidal proteins produced by 5422Bt1 and 5422CBCL, and discuss them with the results obtained because these two Bt corns showed different responses. By doing this, the conclusion can be more supportive.
Although authors used two Bt corns, there is not much discussion/explanation on 5422Bt1 compared to 5422CBCL.
There are 14 figures! Reduce them by combining leaf and root data on the same enzymes/chemicals. For instance, Figure 1 and 8 can be together. If authors do not like this idea, this reviewer suggests use tables.
Round 2
Reviewer 2 Report
Although the manuscript was improved, there are still certain aspects that need to be improved.
Line 116: Please replace desense by defense
Line 365: Measured at 725nm in a UV spectrophotometer? or UV-Vis? 725nm is in the UV range?
There are several formatting errors in the text , both in different types of letters (for example, lines 116, 365), and in not making separations in sections 2.1.1 and 2.1.2 (line 100), (4, 2 and 4.3 are missing inside 4.1. All in italic?). Check the formatting of all the text.
The figures should not be in a separate section (2.3), but between the results.
